# Addressing the Need for a Specialized Disconnection Device in Catheter Connection Management: A Case Study of User-Centered Medical Device Innovation

**DOI:** 10.3390/nursrep15020036

**Published:** 2025-01-24

**Authors:** Amy C. Cole, Nicole Wiley, Kerri Dalton, Daniel R. Richardson, Deborah Allen, Nancy Havill, Lukasz Mazur

**Affiliations:** 1Division of Healthcare Engineering, Department of Radiation Oncology, University of North Carolina at Chapel Hill, Chapel Hill, NC 27599, USA; lukasz_mazur@med.unc.edu; 2Carolina Health Informatics Program, University of North Carolina at Chapel Hill, Chapel Hill, NC 27599, USA; 3Fast TraCS, NC TraCS Institute, University of North Carolina at Chapel Hill, Chapel Hill, NC 27599, USA; nicole_wiley@med.unc.edu; 4Duke University Health System, Durham, NC 27710, USA; kerri.dalton@duke.edu (K.D.); hutch.allen@duke.edu (D.A.); 5Lineberger Comprehensive Cancer Center, University of North Carolina, Chapel Hill, NC 27599, USA; daniel_richardson@med.unc.edu; 6Center for Nursing Excellence, UNC Health, Chapel Hill, NC 27599, USA; nancy.havill@unchealth.unc.edu

**Keywords:** catheters, patient safety, user-centered design, usability, socio-technical, nursing practice workarounds, quality improvement

## Abstract

**Background/Objectives**: Improvements in catheter connection design intended to increase safety have resulted in connections that are difficult to release manually. No medical device exists to safely disconnect catheter connections. Nurses and other users have developed workarounds including use of hemostats, tourniquets, and wrenches. These workarounds are not always successful for performing this task and can break catheters and catheter connections. This study aimed to evaluate a disconnection device to safely disconnect catheter connections. **Methods**: This is a mixed-methods study using a user-centered design approach with triangulation of quantitative and qualitative data mapped to Valdez’s sociotechnical framework. Nurses (*N* = 139) from units across two academic medical centers encompassing diverse patient populations engaged in usability testing and surveys. Data about users’ past catheter disconnection experiences and usability of the specialized disconnection device were collected and analyzed. Triangulation of quantitative data and qualitative themes was mapped using Valdez’s socio-technical framework to complement and strengthen the final design generated for nurses’ user requirements. **Results**: Ninety-five percent of nurses reported previous difficulty with disconnecting luer connections; 93% of those reporting difficulty improvised with readily available medical devices or products to better grip the connected parts. Over 85% of nurses reported positive experiences using the specialized disconnection device; others suggested design improvements for better performance. **Conclusions**: The nurses who tested the developed disconnection device reported high acceptability, accessibility, ease of use, and improved task performance. Moreover, as workarounds develop at points of practice where no systematic solution exists, aiming product development activities at these points help close gaps in achieving and maintaining patient safety. This study was not registered.

## 1. Introduction

Every year in the United States, millions of patients receive prescribed treatments administered through, or removal/irrigation of bodily fluids extracted from, various catheters (e.g., central venous catheters, feeding tubes, drainage tubes, urinary catheters, or other intravenous administration sets) [1,2,3,4]. Catheter connection management is commonly performed by nurses; however, patients and caregivers also manage treatments and/or replace feeding or drainage tubes, at home, if daily or long-term care is necessary [5,6,7,8,9,10,11]. Catheter connection management can be stressful, as these patients are often critically ill and at an increased risk of healthcare-associated infections (HAIs), such as central line-associated bloodstream infections (CLABSI) or catheter-associated urinary tract infections (CAUTI) [6,11,12,13]. Indeed, in the United States, on any given day, one in thirty-one hospital patients will have an HAI, placing these patients at an increased risk of mortality [12,13,14]. 

Scheduled changing of catheter connections (e.g., medical tubing, needleless claves, catheter bags) as well as performing disinfection techniques have been shown to decrease contamination risk [11,15,16,17,18]. While these techniques are effective at decreasing contamination risks, additional challenges exist when the catheter’s “luer” connection seizes between the male and female ends, making it difficult, if not impossible, to disconnect the male and female ends from each other manually (i.e., without the aid of medical devices or products). In a recent pilot study, 90% of patients and caregivers and 79% of nurses reported having experienced challenges with removing needleless claves from the end of central line lumen hubs [19]. 

In recent decades, the luer connection [11,20] has advanced the performance of managing catheters by providing a secure connection among various catheter connection types (e.g., medical tubing, needleless syringes, needleless claves, administration/irrigation bags). However, this advancement has also increased the challenges for end users (e.g., patients, caregivers, and nurses) in loosening such connections after the administration of treatments [19]. For example, the tighter seal and enhanced connection were designed to improve the management of catheters but have specifically made it harder to unscrew them, as they require more force to disconnect. Medical instruments (e.g., hemostats, scissor-like occluding clamps, tourniquets) and non-medical supplies (e.g., general purpose wrenches) have been used as workarounds. The negative repercussions of these workarounds include damage to catheters and catheter connections, requiring patients to undergo catheter repair or replacement procedures. End-users’ preferences are to have a specialized disconnection device to mitigate these challenges [19]. At present, there are limited solutions that address this problem, and current literature and guidelines regarding the challenges and solutions are sparse. 

The purpose of this study was to evaluate a disconnection device to safely disconnect catheter connections through a user-centered design approach and a socio-technical framework to better understand users’ needs for improving the design. 

## 2. Materials and Methods

This study focused on users and their tasks, usability testing to understand users’ perceptions, and an iterative design process [21]. To achieve this, nurses’ previous experiences with luer connections along with their perspectives on a physical prototype of the disconnection device were assessed, mapped to Valdez’s socio-technical framework [22] to understand the interconnections between nurses, their tasks, the technology, and the organizational structure [22], to inform the final design.

### 2.1. Participants

Nurses were recruited from two academic medical centers (AMCs) between September 2023 and June 2024 to evaluate and make iterative improvements to a proof-of-concept prototype of the disconnection device. Approval was obtained from the Institutional Review Boards at both academic medical centers for this study (see Institutional Review Board Statement). Participants verbally consented at AMC 1 and were exempt from consent at AMC 2. Participants were first recruited from AMC 1, which resulted in iterative improvements to the prototype design. This was followed by recruitment at AMC 2. See Figure 1 for study design schema.

Purposive sampling was used at both AMCs to recruit nurses working in units or clinics that care for patients with catheters. Eligibility at both AMCs was based on nursing qualifications, specifically those trained in catheter management care. Nurses without prior experience caring for patients with catheters were excluded. Participants were contacted through email by our research team for scheduling usability sessions in the nurses’ home unit at each respective institution. To avoid introducing bias or persuasion, there was no direct reporting relationship between the research team and the participants. A total of 139 registered nurses (RNs) across both AMCs engaged in usability testing and were then asked to complete a previous experience survey. Participants in usability testing sessions from AMC 2 were also asked to complete a post-usability survey to provide perspectives using the disconnection device. Participants were provided with a QR code to access both the previous experience and post-usability surveys. All surveys were completed within 1 week of the usability testing session. Units included critical care, rehabilitation, adult and pediatric in-patient, blood and marrow transplant, oncology infusion clinics, dialysis, and intensive and progressive care units at both respective AMCs. All participants were observed by researchers while performing the task during the usability testing session. Observational data were documented for qualitative analysis. 

### 2.2. Study Design

A mixed-methods study was designed using a user-centered design approach, including understanding the (i) end-users’ previous experiences with luer connections for context of use and (ii) user perspectives and requirements for product design [21,23,24]. This approach involved collecting both qualitative (usability testing sessions and open-ended survey questions) and quantitative (surveys on previous experiences and post-usability feedback on using the disconnection device) data from nurses to better understand the task of disconnecting luer connections. See Figure 1 for study schema. 

The initial prototype of the disconnection device used in this study was developed based on the collected data from our pilot study [19]. Usability testing sessions were first conducted at an academic medical center (hereafter referred to as AMC 1), where nurses provided feedback on the initial prototype of the disconnection device. When relevant design improvements were uncovered during usability sessions at AMC 1, refinements were made for subsequent testing sessions. Usability sessions were then held at another academic medical center (hereafter referred to as AMC 2) with the revised design (Figure 2) to gain insights specific to usability, length, weight and comfort of the disconnection device. 

#### 2.2.1. Prototype Development

Initial usability testing at AMC 1 involved nascent prototypes, in which iterative improvements were made based on feedback from nurses. This included lengthening the arms of the disconnection device, adjusting the grip diameter, and providing guidance for hand placement. Later usability testing sessions provided additional design improvements including the types of material used for manufacturing and the placement of grooves within the grip area.

##### Design Requirements

The disconnection device is designed to facilitate the unmet need for improved assistance with safely disconnecting various catheter luer connections. The primary design requirements are the ability to securely grip luer connections of varying sizes and to provide the leverage needed to loosen/tighten the connection without causing damage. The resulting design is a symmetric, U-shaped disconnection device measuring 12.5 cm long and 1 cm wide, consisting of two arms connected at one end and open at the other (see Figure 2 and Patents). The disconnection device’s arms remain open at rest to quickly allow for catheter connections to be positioned in the appropriate grip region. The outermost surface of the two arms includes a ribbed area to assist the user in firmly grasping the body of the disconnection device and applying the leverage needed for disconnection. The two main grip sizes represent the median diameter of the most common sizes for various catheter connections and catheter lumen hubs (i.e., external end points of the catheter). Grip area 1, the largest grip area, is 11 mm in diameter and located closest to the disconnection device opening which will accommodate various connections for catheter connection management (e.g., needleless syringes, needleless claves, administration/irrigation bags). Grip area 2, the smaller grip area, is 5 mm in diameter and will accommodate CVC lumen hubs, PICC lumen hubs, and other IV-lumen hubs. 

##### Material Selection and Prototyping

The body of the disconnection device is constructed from plastic, which is lighter weight and has a lower risk of damaging hubs than metal medical devices. Elastomeric portions line each side of the hub grip areas to aid in securely gripping the catheter connections. The prototype of the disconnection device was designed in 3D CAD modeling software Onshape version 1.177, and the physical prototype of the disconnection device was manufactured on a Formlabs Form 2 3D printer with Tough 1500 and Elastic 50A material resin (https://formlabs.com/) for the body and grip areas, respectively. The processes described were solely for usability testing purposes, and the final disconnection device will be manufactured following stringent quality and control standards. 

##### Disconnection Device Operation

To facilitate disconnection, one or two disconnection devices may be used. Prior to use, the disconnection device should be wiped with a sanitizing cloth. For disconnection, the user holds the device in either hand, guiding the catheter or catheter connection to the appropriately sized grip area. By squeezing the arms of the device together, the user can firmly grasp the catheter or catheter connection. In instances when additional leverage is needed, two disconnection devices can be used simultaneously. For example, if the first device is applied to the catheter, the second device is applied to the catheter connection. Leverage is then applied in opposing directions for disconnection. Once this is achieved, the user relaxes their grip and removes the disconnection device from the male or female part of the catheter or catheter connection. The disconnection device should be wiped with a sanitizing cloth after each use.

#### 2.2.2. Previous Experience Survey

Our research team developed the survey questions to validate evidence from our pilot study, which indicated users had challenges with disconnections. All participants were informed that the survey focused solely on experiences with manually disconnecting catheter connections. 

Quantitative: Nurses at AMC 1 reported on the frequency at which problems occur with disconnections using a Likert-type scale, with responses ranging from 0 times per month to 1–4 times per month, 5–10 times per month, and more than 10 times per month. Nurses at AMC 1 also reported on whether they had received training related to mitigating problems with disconnecting luer connections, with a yes/no response option.

AMC 1 and AMC 2 nurses’ previous experiences disconnecting luer connections were quantified through three survey questions, including whether they had experienced any of the following: (1) difficulty disconnecting, (2) request for help with disconnecting, or (3) use of an assistive medical device/product to disconnect. 

Qualitative: Nurses at AMC 1 reported the types of medical devices/products currently utilized with luer connections and responded to an open-ended question to further elaborate on their previous experiences with luer connections. This open-ended question was only included in the survey from AMC 1, as we recognized this information was verbalized during usability testing sessions; therefore, it was removed from the survey to reduce participant survey burden.

#### 2.2.3. Usability Testing

Nurses’ perspectives of the disconnection device were assessed through usability testing at AMC 1 and AMC 2. Usability testing sessions were designed to identify aspects of the disconnection device that could be refined or enhanced for informing the final design decisions [25,26]. To conduct usability testing efficiently across units, a transportable kit was created containing varying luer connections, such as central lines and needleless connectors, tubing connections, syringe connections, and feeding tube connections. Usability testing sessions were either held in workrooms, training rooms, or conference rooms of each respective unit. When testing sessions were held in “training rooms”, mannequins were used in addition to the transportable kit. Sessions were held in 29 units (AMC 1, *N* = 11; AMC 2, *N* = 18), with participant counts ranging from 2 to 10 nurses per session.

A multidisciplinary research team including a clinical nurse scientist, a registered nurse, a biomedical engineer, and a human factors expert led usability testing sessions. At AMC 1, sessions were led by our human factors expert and biomedical engineer. At AMC 2, sessions were led by our clinical nurse scientist and registered nurse. Our research team began each usability testing session by providing participants with background information on the development of the disconnection device. Nurses were not given instructions on how to hold the disconnection device, as our research team was interested in observing initial responses on how the disconnection device best fit in their hands, as well as how nurses would use the disconnection device on various connections. During usability sessions, nurses were asked to perform the task of disconnecting varying luer connections using the disconnection device. Nurses were also asked to describe their current processes for disconnecting luer connections. We documented both non-verbal signs (e.g., hand placement, catheter placement, and performance success and failure) from observing nurses’ task performance, as well as their verbal feedback on both current processes (e.g., challenges, successes, and workarounds) and use of the specialized disconnection device (e.g., challenges and successes, suggested improvements, and satisfaction).

#### 2.2.4. Post Usability Survey

Our research team developed an 11-question survey, including Likert scale and open-ended questions, to obtain evidence on the user experience of the specialized disconnection device. Post-usability surveys were only conducted at AMC 2. This contrasts with AMC 1, in which we solely focused on observational data aimed at uncovering initial design solutions for a high-fidelity prototype of the disconnection device.

Quantitative: Post-usability surveys at AMC 2 were conducted to quantify the usability, length, weight, comfort, grip size, grip function, and acceptability of the disconnection device using a 5-point Likert scale. Acceptability included two questions, “How likely are you to use the device yourself” and “How likely are you to recommend others use this device”.

Qualitative: Nurses at AMC 2 reported on their experience using the disconnection device after usability testing. Open-ended survey questions included “What additional features or modifications would enhance the usability of the device?”; “Which aspects of the device functioned well?”; and “Are there any concerns with how this device functions?”。

#### 2.2.5. Triangulation Using Socio-Technical Framework

The quantitative data from surveys were triangulated with the qualitative themes generated from both usability sessions and open-ended survey questions to gain a comprehensive understanding of nurses’ prior experiences as well as perceptions of a specialized disconnection device [27]. These data were then mapped using Valdez’s socio-technical framework to complement and strengthen the understanding of nurses’ user requirements within an organizational structure for finalizing the design of the disconnection device [22]. Valdez’s socio-technical framework was developed to understand the complexity and interconnections between four domains: users, tasks, organizational structure, and technology (See Figure 1 for Valdez’s socio-technical framework and Table 1 for mapping of data collection to Valdez’s framework ). The user domain considers human factors and context, including users’ skills, perceptions, and motivation. The task domain focuses on urgency, habituation, and strategies for task completion. The organizational structure domain considers cultural norms, standard operating procedures, and communication channels. The technology domain focuses both on computer applications and how tasks, skills, and communication can influence the success of the organization. This study evaluated these interconnections but associated technology with physical devices and products rather than computer applications. This study applied an adapted socio-technical framework for addressing nurses’ need for a specialized disconnection device in catheter connection management. 

### 2.3. Data Analysis

All data were subjectively assessed, analyzed, and quantified from survey responses and usability testing sessions. To reduce unintentional bias of results, further review was conducted by two members of our research team (DRR and LM), who did not have prior experience in performing this task. Data collected from surveys and usability testing sessions were first mapped to Valdez’s framework, as displayed in Table 1, to establish a priori codes. Qualitative data was analyzed as described in Section 2.3.3 and Section 2.3.4 to generate themes. Themes generated from the qualitative data were then combined with the quantitative data to determine how each theme best fit under one of the four domains, while highlighting the interconnections between each of the four domains: user, task, organization, and technology. 

#### 2.3.1. Participants

Descriptive statistics were used to summarize the number of years nurses have been performing tasks related to disconnecting luer connections and the units in which nurses are performing this task.

#### 2.3.2. Previous Experience Survey—Quantitative

Descriptive statistics were used to summarize nurses’ previous experiences disconnecting luer connections. Each aspect (i.e., trouble disconnecting, requesting help, and use of assistive medical device) was counted and analyzed through frequency tables, revealing the distribution among nurses. 

#### 2.3.3. Previous Experience Survey—Qualitative

Open-ended survey responses were analyzed using a hybrid approach to thematic analysis, including three phases of analysis in which data were refined to assess the meaningfulness of themes related to nurses’ previous experiences with luer connections [28]. Initial responses were first mapped to Valdez’s framework. Then, in phase 1 of the thematic analysis, data from survey responses were categorized by a priori themes based on user-centered design principles, such as current tasks, workflows, usability, and functionality. In phase 2, we created the initial a posteriori codes, and in phase 3, we combined the a priori and a posteriori codes into family codes to structure the findings from the open-ended survey responses [28]. All codes underwent a comprehensive review and were systematically clustered into overarching themes through collaborative discussion and consensus-building among the research team. 

#### 2.3.4. Usability Testing and Post Usability Survey—Qualitative

Usability testing session data and open-ended post-usability survey responses were analyzed using a hybrid approach to thematic analysis, including three phases of analysis in which data were refined to assess the meaningfulness of themes related to nurses’ perspectives using the disconnection device [28]. Thematic analysis was performed as described in Section 2.3.3. 

#### 2.3.5. Post Usability Survey—Quantitative

Post usability surveys were only collected at AMC 2. Descriptive statistics were used to summarize perspectives with disconnecting luer connections while using the disconnection device. Each aspect (i.e., usability, length, weight, comfort, grip size, grip function, and acceptability) was counted and analyzed through frequency tables, revealing the distribution among nurses. 

#### 2.3.6. Triangulation 

Following Valdez’s socio-technical framework [22], all data underwent a comprehensive review and were systematically clustered into themes to display the interconnection between four domains, (i) nurses, (ii) organizational structure, (iii) tasks, and (iv) technology (or medical devices/products). 

## 3. Results

### 3.1. Participants

Nurses at AMC 1 engaged in usability testing sessions (*n* = 62) and completion of previous experience surveys (*n* = 61) from September 2023 to November 2023. Nurses at AMC 2 engaged in usability testing sessions (*n* = 77), completion of previous experience surveys, and post-usability surveys (*n* = 66) from April 2024 to June 2024. Nurses who completed the surveys self-reported years of experience, as displayed Figure 3.

### 3.2. Previous Experience Survey—Quantitative

Ninety-five percent (*n* = 124/131) of nurses reported previous difficulty with disconnecting luer connections. Of the 124 nurses reporting difficulties, 75% (AMC 1, *n* = 41/59; AMC 2, *n* = 52/65) have requested assistance, and 93% (AMC 1, *n* = 51/59; AMC 2, *n* = 64/65) have applied a medical device or wrapped medical products around the connection points to gain a better grip for disconnection.

Data collected from AMC 1 also included the frequency in which this occurs and whether they received training on how to resolve this problem. Of the nurses who reported experiencing problems (*n* = 59/61), 78% (*n* = 46/59) reported that this occurs one to four times per month, 20% (*n* = 12/59) five to ten times per month, and 2% (*n* = 1/59) more than ten times per month. Only 23% (*n* = 11/61) of nurses reported receiving training on how to resolve problems with luer connections.

### 3.3. Previous Experience Survey—Qualitative

Sentiments from the previous experience survey focused around four main themes: patient safety and infection control, nurses’ time, challenges and workarounds, and undesirable task performance (Table 2). Patient safety and infection control focused on the (i) risk of infection due to improper handling of various catheters, (ii) importance of following protocols to prevent infection, and (iii) impact on patient safety from using off-label medical devices such as hemostats. Nurses’ time and efficiency focused on time-consuming processes to mitigate this problem. Challenges and workarounds focused on the various methods and devices to mitigate disconnection challenges. Undesirable task performance focused on (i) the inconsistency in disconnections and (ii) lack of training. Nurses reported relying on using off-label medical devices (e.g., hemostats or clamps) or wrapping medical/non-medical products (e.g., tourniquets, medical gloves, paper tape, gauze, alcohol pads, Coban, Kerlix, and dry paper towels) around the connection points to gain a better grip when performing this task.

### 3.4. Usability Testing and Post Usability Surveys—Qualitative 

The observations and sentiments discussed during the usability testing sessions along with the open-ended responses from the post-usability survey focused around six main themes: usability and ease of use, design functionality, design improvements, infection prevention, accessibility, and acceptability (Table 3). During usability testing sessions, nurses acknowledged that their unit (e.g., oncology infusion, bone marrow and transplant, critical care) no longer allowed hemostats for this use case, due to previous reports of hemostats causing damage to central lines. Nurses repeatedly highlighted the need for improving strategies for disconnecting luer connections.

### 3.5. Post Usability Survey—Quantitative

Table 4 summarizes nurses’ perspectives using the disconnection device during usability testing at AMC 2. The majority of nurses reported either being satisfied or very satisfied with the usability, length, weight, and comfort of the disconnection device. The majority of nurses also reported high acceptability of the disconnection device, indicating the grips are the correct size, the disconnection device worked as intended, and that nurses would either use the disconnection device themselves or recommend it to others. 

### 3.6. Triangulation

Based on the mapping of data to Valdez’s socio-technical framework (Table 1), Figure 4 displays how the themes from the qualitative data were combined with the quantitative data to highlight the interconnection between the four socio-technical domains, (i) user (nurses), (ii) tasks, (iii) organizational structure, and (iv) technology (or medical devices/products).

Usability testing and surveys resulted in developing a better understanding of challenges nurses face in performing this task. Results indicate that nurses rely on workarounds when they are unable to manually disconnect the catheter connection. When trying to mitigate these challenges, the design elements they seek in technology (e.g., medical devices and products such as hemostats, clamps, and tourniquets) focus on usability, ease of use, and functionality. Nurses reported on the frequency in which this challenge occurs and that performing this task with current workarounds is undesirable. Nurses’ performance of these tasks and the technology used are interconnected to the organizational structure, as it relates to patient safety, infection control, nurses’ time spent performing this task, and lack of training on how to safely mitigate the challenges faced when performing the task. When nurses were presented with the specialized disconnection device for luer connections at AMC 2, they reported high acceptability of the proposed disconnection device. Nurses also reported that to improve task performance, a device needs to be accessible. 

## 4. Discussion

In this study, nurses from two large academic medical centers engaged in usability testing and completed surveys to provide perspectives on previous experiences with catheter connection management, as well as perspectives on using a specialized disconnection device for disconnecting luer connections. The purpose of this study was to evaluate a specialized disconnection device for disconnecting luer connections. Specifically, we focused on the interconnection between nurses, their tasks, the structure of the organization, and the devices and products available to perform the task. 

Evidence from the usability sessions and surveys indicates that problems with disconnecting luer connections are prevalent across multiple units. When problems arise, nurses improvise by using workarounds [29] such as medical devices and products that are immediately accessible. This includes medical devices such as hemostats and clamps, which can provide leverage but are more likely to cause a medical error and negatively impact patient safety outcomes. Other medical products such as tourniquets, compression wraps, and alcohol pads are used to provide friction around the connections but are less effective for disconnection. While these particular workarounds have not previously been studied, a recent systematic review in critical care nursing reported that workarounds are common in nursing practice [30]. Workarounds in healthcare settings are often performed because there is a culture that supports resiliency and resourcefulness to continue providing care to patients, despite any obstacles encountered [29]. While workarounds can mitigate a dysfunctional process, they leave the underlying problem unresolved, increase the opportunity for medical errors to occur [30,31,32], and are also linked to poor patient safety outcomes [29]. Unfortunately, evidence from this study suggests that medical errors have occurred from these workarounds. For example, hemostats were reported to have caused damage to catheters, resulting in the need for repair or replacement of catheters, impacting patient safety and overall healthcare costs. Currently, no device exists to safely disconnect catheter connections; therefore, those performing this task must decide between unsafe workarounds or placing patients at higher risk of infection by not changing catheter connections.

In high-reliability organizations, recognition of systematic problems is important for developing solutions [33]. Evidence from this study suggests there is a systematic problem in disconnecting luer connections and the need for developing a solution to improve patient safety. The American Hospital Association’s 2024 goals for patient safety include developing improvement tools and decreasing preventable harms [34]. This study used a user-centered design approach to ensure that nurses’ perspectives were incorporated into the design of a specialized disconnection device to address this systematic problem. Our approach is consistent with previous studies that report engaging a diverse group of end-users in the design process increases the likelihood for addressing the end-users’ needs and tailoring devices to their specific task, therefore reducing the risk for medical errors and the development of safer products [21]. Evidence from this study supports those notions, as nurses’ feedback was specific to how best to tailor this disconnection device to meet their specific needs for disconnecting luer connections and ensuring the disconnection device is designed for easy cleaning, the potential for single patient use, and sterile packaging options. 

Evidence from the usability testing sessions at AMC 2 demonstrates that nurses are likely to use this specialized disconnection device and would also recommend it to others. Most nurses found the disconnection device easy to use and functional and that the size and comfort were correct. Nurses reported high acceptability of this disconnection device and consistently asked when they would have access to the disconnection device in their units. Some nurses suggested design improvements before the device would meet their needs. For example, nurses suggested improving material design to ensure durability, variable sizes to apply torque, and ridges within grips for better performance. Providing nurses with access to a specialized disconnection device needs to be supported at the organization level. Evidence from this study indicates several organizational factors should be addressed, including the connection between nurses’ current performance of this task and how this impacts risks to patient safety and infection control, increased time spent performing the task or aiding others in performing the task, and the lack of training they have been provided on how to successfully mitigate this systematic problem. When leadership commits to a culture of safety by listening to concerns and deploying process improvement tools, they position themselves to generate and achieve higher levels of patient safety and become recognized as high-reliability organizations [33]. Hospitals should consider the evidence from this study as the first step in understanding the systematic problem nurses are facing with catheter connection management. Considerations should be made for the utilization of this specialized disconnection device in clinical settings and adjacent training to assess the impact this has on reducing workarounds as well as associated healthcare costs and improvement in patient safety outcomes.

### Strengths and Limitations

This study was a subjective assessment of nurses’ experiences; therefore, to mitigate unintentional biases of data collection, analysis, and reporting, the usability sessions were performed by differing research team members, surveys were used to collect data, and collective consensus was reached on generated themes and final analysis. Further review of results was conducted by two members of our research team (DRR and LM) who did not have prior experience in performing this task. This study was conducted at two academic medical centers; therefore, results may not be generalizable to other settings, such as smaller community center hospitals, where user workflows may be different. However, our study had broad representation across multiple clinical units among nurses with varying years of experience with catheter connection management, which may be more applicable than differences among healthcare institutions. This study is limited by not observing nurses performing this task in clinical settings with the medical devices and products currently available. We were also limited by not testing the specialized disconnection device in a clinical setting, as the disconnection device had not been registered with the FDA at the time of this study. However, conducting usability testing sessions allowed nurses to discuss the utilization of current medical devices and products and perform the task in a simulated environment without burdening patients. The insights obtained ensured the design of the disconnection device met their needs and supported their current workflows. Additionally, by using both a user-centered design approach and Valdez’s socio-technical framework, we have the evidence to strengthen the final design of the disconnection device by understanding the interconnection of nurses’ needs and how the organizational structure impacts nurses’ user requirements [22]. A further limitation is that nurses recruited for this study were those with previous experiences with managing catheter connections, which may not reflect all potential users. However, this was also viewed as a strength, as nurses had the necessary experience to provide more meaningful feedback on user requirements. Future studies will include evaluating current costs associated with identified workarounds and the damage they may cause to catheters, as well as the performance of the final design of the disconnection device in real-world clinical settings. 

## 5. Conclusions

The nurses who tested the developed disconnection device reported high acceptability, accessibility, ease of use, and improved task performance. By applying a socio-technical framework, we were also able to understand the context around nurses’ performing the task of disconnecting luer connections. Using this approach and framework, this study identified a systematic problem with disconnecting luer connections. This approach and framework also ensured the design of the disconnection device met both the nurses’ functional and workflow needs but also accounted for the interconnection between nurses and their organizational structures. This approach provides a foundation for optimizing organizational success by acknowledging a systematic problem, designing a user-centered disconnection device to mitigate the problem, and suggesting that implementation of the device can improve patient safety outcomes and task performance for nurses.

## 6. Patents

The disconnection device described in this manuscript is protected under intellectual property laws and secured by patents, ensuring exclusive rights to its use and development. 

## Figures and Tables

**Figure 1 nursrep-15-00036-f001:**
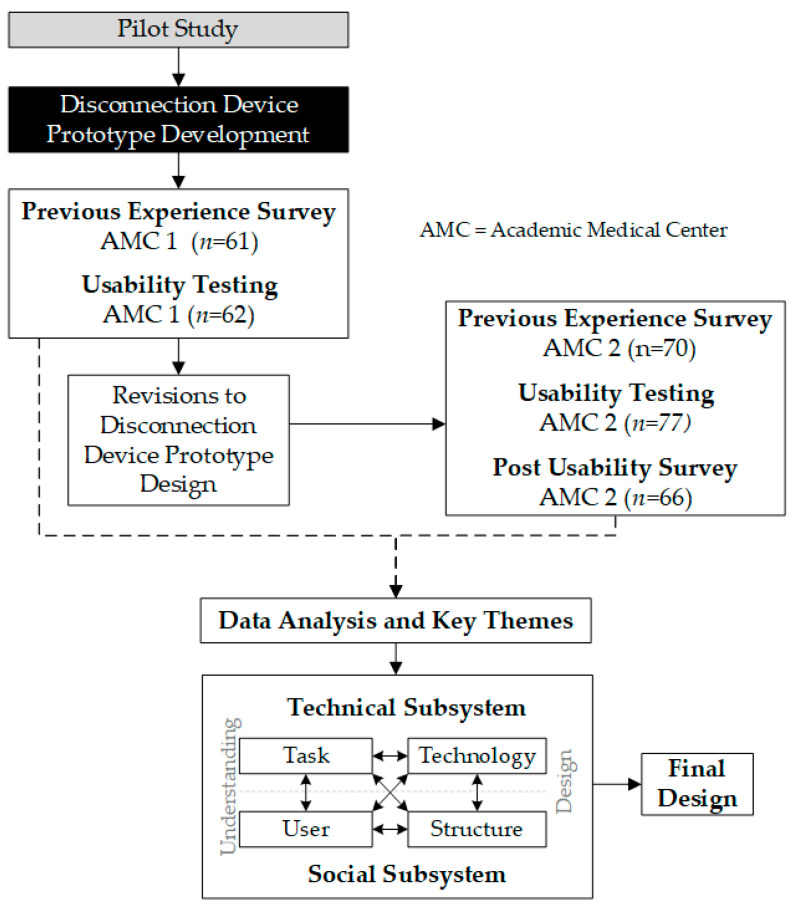
Study schema—includes pilot study [19] and Valdez’s socio-technical framework, Reproduced with permission from [22].

**Figure 2 nursrep-15-00036-f002:**
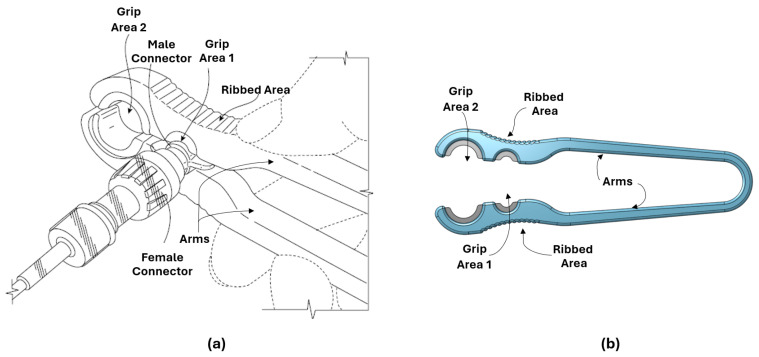
(**a**) Disconnection device schematic—hand placement is intended within the ribbed area; however, hands are shown in alternate location to visually display the ribbed area. (**b**) Side view of disconnection device.

**Figure 3 nursrep-15-00036-f003:**
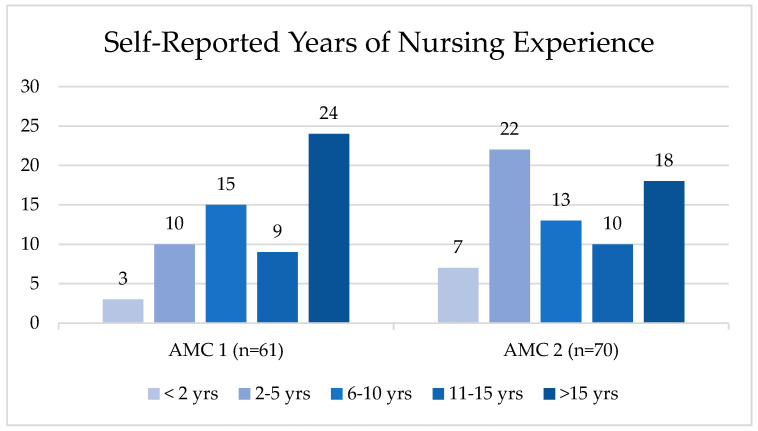
Nurses’ self-reported years of nursing experience.

**Figure 4 nursrep-15-00036-f004:**
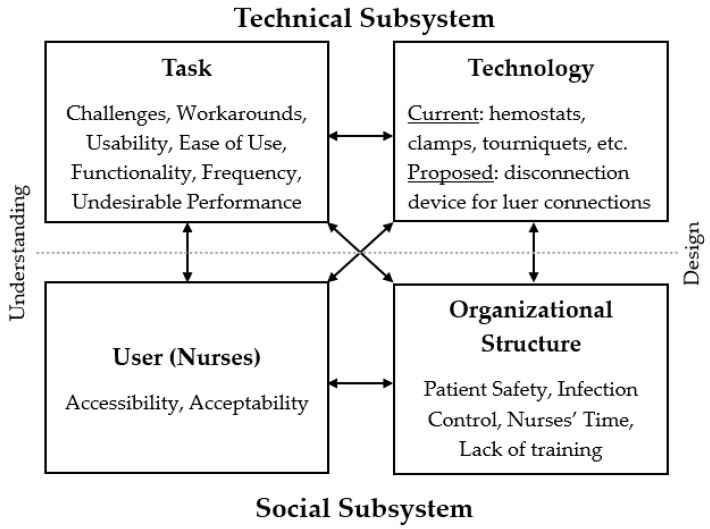
Adapted (Valdez’s) socio-technical framework—addresses need for specialized disconnection device. Adapted with permission from [22].

**Table 1 nursrep-15-00036-t001:** Data collection mapped to Valdez’s socio-technical framework.

	User	Task	Organization	Technology
**Previous Experience Survey**				
Nursing years of experience	X	X		
Difficulty disconnecting	X	X		X
Requested help with disconnecting	X	X	X	X
Used a medical device/product to aide in disconnecting	X	X		X
Training	X	X	X	X
Frequency of occurrence	X	X	X	X
Current workarounds	X	X	X	X
**Post Usability Survey**				
Usability	X			X
Length	X			X
Weight	X			X
Comfort	X			X
Acceptability	X			X
What additional features or modifications would enhance the usability of the device?	X			X
Which aspects of the device functioned well?	X			X
Are there any concerns with how this device functions?	X	X		X
**Usability Testing Sessions**				
Feedback on current processes/challenges	X	X	X	X
Observation of task performance	X	X		X
Current workarounds	X	X	X	X
Suggested improvements to specialized disconnection device	X	X		X
Satisfaction with specialized disconnection device	X	X		X

**Table 2 nursrep-15-00036-t002:** Qualitative themes from previous experience surveys.

Theme	Example Quotes Representative of Theme
Patient Safety and Infection Control	“If I’m struggling to remove the clave from the hub, I first try using gloves to improve grip. If that fails, I try hemostats, I get nervous that the hemostats will break the lumen hub. I’ve seen this happen to peer before.” “Hemostats are against protocol to use in our department.” “I have used hemostats or tourniquets or both to get it to come off, though I did stop using the hemostats after we realized they were often cracking the line.” “I have had claves break or not be changed because we couldn’t get it to release.” “This is a real problem. Not only with claves of CVADs. But also disconnecting tubing from NG tubes. CVAD infection risk is a big concern and is the primary focus of the children’s hospital quality work this year.”
Nurses’ Time	“If the hemostats do not work, I reach out for help from other nurses.” “When it’s difficult to remove it takes a lot of time.” “It can be stressful and time consuming when claves are stuck to lumen hubs.”
Challenges and Workarounds	“I usually use hemostats to remove them. Rarely do they come off without.” “Rare that I have to get a tool but not rare that I struggle” “We often have to use a hemostat 2 times to help remove the clave.” “We also experience this issue when placing a line. Sometimes we may connect the flush a little too tight and it is difficult to get off.” “Often, will open an entire CVC kit just to get the scissor clamps to remove a clave…very wasteful.” “Our unit has many many central lines and when performing protocoled line change every 4 days, the claves are always really tough to remove.”
Undesirable Task Performance	“It occasionally occurs that it is difficult to remove a clave from the hub but using a hemostat has always worked for me to get the clave removed.” “We mostly place the IVs and PICCs, so we are not usually changing tubing or claves accept when troubleshooting lines. We do experience this issue from time to time. We use a glove, hemostats, tourniquet or simply ask another team member to come and assist with removal.” “Usually, I can get it off without using a device but if it is "stuck" I will use 2 pairs of hemostats to get a better grip.” “I have never been educated on what to do when this problem arises.”

**Table 3 nursrep-15-00036-t003:** Qualitative themes from usability testing sessions and open-ended survey responses.

Theme	Example Quotes Representative of Theme
Usability Ease of Use	“[I am] able to stabilize the valve cap and turn the cap to disconnect it.” “To stabilize the line, it works great. Can also see how using two of the devices would be helpful in that instance.” “Good size to hold in large/small hands, easy to use.” “The silicone grippers on the end worked well.”
Design Functionality	“You get more torque with this device than with use the plastic cap and tourniquet.” “The grip worked well, and I was able to pinch the device around the cap.” “The size works well, and I think it did a good job of helping to get the end caps off.” “Our dialysis lines can get slippery and can get sucked into each other. We need a strong device with adequate grip.”
Design Improvements	“The device could use a third hole that is the same size and the smaller one to reapply caps.” “I think a silicone with ridges may be more effective in gripping.” “I would try to add some sort of teeth/groves into the grip, so the cap just doesn’t twist in the grip.” “I feel the material should be sturdier, the plastic seemed flimsy, too much bend in it.” “Variable sizes—ability to apply torque.”
Infection Prevention	“Sterile device for sterile line changes.” “Was wondering if there was a disposable sterile option we could drop if needed during a sterile procedure?” “Need more clarification regarding single use, ability to reuse, sanitation, etc.”
Accessibility	“Would like this to be stocked on our unit.” “Could it be smaller for ease of carrying in scrub pocket?” “The smaller grip thing is very useful, I would like the option of having a device that is smaller, say that I could put on my badge.” “I have arthritis in my thumbs, and this would make a huge improvement in my ability to change caps”
Acceptability	“This is awesome!” “I could see many nurses, myself included finding frequent use in a device like this. Very exciting.” “Wish I had thought to design this.”

**Table 4 nursrep-15-00036-t004:** AMC 2—nurses’ perspectives using the proposed disconnection device.

	AMC 2 (*N* = 66)
Usability	
Very dissatisfied	0% (*n* = 0/66)
Dissatisfied	5% (*n* = 3/66)
Neutral	12% (*n* = 8/66)
Satisfied	44% (*n* = 29/66)
Very Satisfied	39% (*n* = 26/66)
Length *	
Very dissatisfied	0% (*n* = 0/65)
Dissatisfied	5% (*n* = 3/65)
Neutral	18% (*n* = 12/65)
Satisfied	46% (*n* = 30/65)
Very Satisfied	31% (*n* = 20/65)
Weight	
Very dissatisfied	0% (*n* = 0/66)
Dissatisfied	3% (*n* = 2/66)
Neutral	11% (*n* = 7/66)
Satisfied	47% (*n* = 31/66)
Very Satisfied	39% (*n* = 26/66)
No Response	
Comfort	
Very dissatisfied	0% (*n* = 0/66)
Dissatisfied	3% (*n* = 2/66)
Neutral	17% (*n* = 11/66)
Satisfied	44% (*n* = 29/66)
Very Satisfied	36% (*n* = 24/66)
Are the grips the correct size for your application? *	
Yes	79% (*n* = 50/63)
No	21% (*n* = 13/63)
Did the device grip the connections as intended? *	
Yes	81% (*n* = 52/64)
No	19% (*n* = 12/64)
Acceptability	
I would use the device myself. *	
Strongly disagree	5% (*n* = 3/65)
Disagree	5% (*n* = 3/65)
Neutral	5% (*n* = 6/65)
Agree	29% (*n* = 19/65)
Strongly agree	57% (*n* = 37/65)
I would recommend the device to others. *	
Strongly disagree	5% (*n* = 3/65)
Disagree	2% (*n* = 1/65)
Neutral	8% (*n* = 5/65)
Agree	23% (*n* = 15/65)
Strongly agree	63% (*n* = 41/65)

* Missing data from respondents, as respondents were not required to answer all questions.

## Data Availability

The raw data supporting the conclusions of this article will be made available by the authors on request.

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
