# Peer review of "Addressing the Need for a Specialized Disconnection Device in Catheter Connection Management: A Case Study of User-Centered Medical Device Innovation"

_nursrep, 2025, doi:10.3390/nursrep15020036_

Round 1

Reviewer 1 Report

Comments and Suggestions for Authors

Please see the attached file for my comments.

Author Response

Thank you for your insightful comments and interest in our research. We appreciate the time and effort you have dedicated in providing valuable feedback on our manuscript.

Comment 1. In Figure 1. Study Schema and Section 2.2 Study Design, the authors mention utilizing Valdez’s socio-technical framework "to inform the final design." However, the manuscript neither presents the final design influenced by the framework (or any other study results) nor provides a discussion on how the framework was applied to shape the device's design. The authors are encouraged to expand and strengthen that section of the manuscript.

Response 1: We agree and have added a table that displays how our data collection was initially mapped to Valdez’ framework. (Table 1. Data collection mapped to Valdez' socio-technical framework). We also added text to section, 2.3 Data Analysis, page 8, lines 277-284, to clarify how the data was analyzed.

All data were subjectively assessed, analyzed and quantified from survey responses and usability testing sessions. To reduce unintentional bias of results, further review was conducted by two members of our research team (DRR and LM), who do not have prior experience in performing this task. Data collected from surveys and usability testing sessions were first mapped to Valdez’ framework, as displayed in Table 1, to establish a priori codes. Qualitative data was analyzed as described in Section 2.3.3 and Section 2.3.4 to generate themes. Themes generated from the qualitative data were then combined with the quantitative data to determine how each theme best fit under one of the four domains, while highlighting the interconnections between each of the four domains; user, task, organization and technology.

We have displayed the qualitative themes in Tables 3 and 4. Figure 4 displays the interconnections between the qualitative and quantitative findings.

We have chosen not to display an image of the final design to protect our intellectual property.

Comment 2. Could the authors elaborate further on the rationale for designing this new medical device? While the introduction provides some context, the data presented later in the study suggests relatively low frequency of use. It would be helpful if the authors could expand on why developing a specially designed tool is worthwhile, as the current evidence may not be sufficient to make a strong case. 

Response 2: Thank you for suggesting we further clarify the rationale for this new device. We have modified this section of the introduction and added a statement that provides a rationale for why this is needed. (Page 2, Introduction Paragraph 3, Lines 66-68 and 73-77)

In recent decades, the luer connection (19) has advanced performance of managing catheters by providing a secure connection among various catheter connection types (e.g., medical tubing, needleless syringes, needleless claves, administration/irrigation bags). However, this advancement has also increased the challenges for end-users (e.g., patients, caregivers, and nurses) in loosening such connections after the administration of treatments (18). For example, the tighter seal and enhanced connection was designed to improve the management of catheters, but has specifically made it harder to unscrew them, as it requires more force to disconnect. Medical instruments (e.g., hemostats, scissor occluding clamps, tourniquets) and non-medical supplies (e.g., general purpose wrenches) have been used as workarounds. The negative repercussions of these workarounds include damage to catheters and catheter components, requiring patients to undergo catheter repair or replacement procedures. End-users’ preferences …

Comment 3. The results described between Lines 267 and 274 would be more effectively presented as a plot for better clarity and visual impact. 

Response 3: Thank you for making this suggestion. We have revised the results section (3.1 Participants, page 9, line 322-329) by adding a bar chart to display the number of years of nursing experience. We have also deleted the corresponding text, as to not be redundant.

Nurses at AMC 1 engaged in usability testing sessions (n=62) and completion of previous experience surveys (n=61) from September 2023 – November 2023. Nurses reported having less than 2 years (n=3), 2-5 years (n=10), 6-10 years (n=15), 11-15 years (n=9), or more than 15 years (n=24) experience. Nurses at AMC 2 engaged in usability testing sessions (n=77), completion of previous experience surveys (n=70) and post-usability surveys (n=66) from April 2024 – June 2024. Nurses reported having less than 2 years (n=7), 2-5 years (n=22), 6-10 years (n=13), 11-15 years (n=10) or more than 15 years (n=18) experience.  Nurses’ self-reported years of experience is displayed in Figure 3.

Comment 4. The authors should review the caption links, as several are broken and display “Error! Reference source not found.” Examples include Lines 97, 105, 129, 210, 276, 293, 303, 310, and 320.

Response 4: Thank you for bringing this to our attention. All links have been fixed.

Reviewer 2 Report

Comments and Suggestions for Authors

Innovative article, highly relevant to the field of nursing, both for presenting the development of a medical device designed to safely disconnect catheter connections, enhancing patient safety, and for using a mixed-methods approach centered on user-centered design with data triangulation based on Valdez's sociotechnical framework. Below are some recommendations for adjusting the article.

Abstract: It is suggested to begin the methods section by stating, "This is a mixed-methods study using a user-centered design approach with triangulation of quantitative and qualitative data mapped to Valdez's sociotechnical framework." Also, include in the methods section the sentence currently found in the results section that specifies the sample size and scope: "Nurses (N=139) from units covering diverse patient populations were evaluated through usability testing and surveys." Specify that the scope included two major academic medical centers in North Carolina. Add to the conclusion: "The nurses who tested the developed medical device reported high acceptability, accessibility, ease of use, and improved task performance."

Introduction: The introduction effectively contextualizes the topic's relevance, supported by 20 references, clearly presents the problem, and concludes with the objective. However, the objective presented in the introduction differs from the one stated in the abstract. The authors should align the objective in the introduction with the one in the abstract. The last two sentences of the final paragraph should be removed from the introduction and included in the materials and methods section.

Materials and Methods: Clarify whether the nurses signed an informed consent form agreeing to participate in the study, as this is not explicitly mentioned. Include the sample size information (currently in the results section) in the methods section and describe how participants were recruited (whether intentional or random), along with the inclusion and exclusion criteria. It is recommended to mention in the methods section that nurses were observed by researchers during task execution, as this detail appears only in the article's strengths and limitations section.

Results: The first paragraph of section 3.1 should be moved to the methods section, and the results section should begin with the second paragraph, describing the profile of the studied sample.

Conclusion: The conclusion should address the research objective. Therefore, the authors should include the following statement: "The nurses who tested the developed medical device reported high acceptability, accessibility, ease of use, and improved task performance."

References: Ensure compliance with the journal's requirement regarding the percentage of references published within the last five years, as many of the 37 references cited are older than five years.

Author Response

Thank you for your insightful comments. We appreciate the time and effort you have dedicated in providing valuable feedback on our manuscript. We believe the incorporated changes that reflect your suggestions are value added for the readers of Nursing Reports.

Comment 1: Abstract: It is suggested to begin the methods section by stating, "This is a mixed-methods study using a user-centered design approach with triangulation of quantitative and qualitative data mapped to Valdez's sociotechnical framework." Also, include in the methods section the sentence currently found in the results section that specifies the sample size and scope: "Nurses (N=139) from units covering diverse patient populations were evaluated through usability testing and surveys." Specify that the scope included two major academic medical centers in North Carolina. Add to the conclusion: "The nurses who tested the developed medical device reported high acceptability, accessibility, ease of use, and improved task performance."

Response 1: We agree and have made the following edits to the abstract. (Page 1, Line 18-39)

Abstract: Background/Objectives: Improvements in catheter connection design intended to increase safety have resulted in connections that are difficult to release manually. No medical device exists to safely disconnect catheter connections. Nurses and other users have developed workarounds including use of hemostats, tourniquets, and wrenches. These workarounds are not always successful for performing this task and can break catheters and catheter connections. This study aimed to evaluate a medical device to safely disconnect catheter connections. Methods: A user-centered design approach utilizing Valdez’ socio-technical framework guided this research. Nurses from two academic medical centers engaged in multiple usability sessions. This is a mixed-methods study using a user-centered design approach with triangulation of quantitative and qualitative data mapped to Valdez's sociotechnical framework. Nurses (N=139) from units across two academic medical centers encompassing diverse patient populations engaged in usability testing and surveys. Data about users’ past catheter disconnection experiences and usability of the specialized medical device were collected and analyzed. Triangulation of quantitative data and qualitative themes was mapped using Valdez’ socio-technical framework to complement and strengthen the final design generated for nurses’ user requirements. Results: Nurses (N=139) from units encompassing diverse patient populations engaged in usability testing and surveys. Ninety-five percent of nurses reported previous difficulty with disconnecting luer connections; 93% of those reporting difficulty improvised with readily available medical devices or products to better grip the connected parts. Over 85% of nurses reported positive experiences using the specialized medical device, others suggested design improvements for better performance. Conclusions: Research guided by user-centered design principles aimed at medical device development results in successful product development. The nurses who tested the developed medical device reported high acceptability, accessibility, ease of use, and improved task performance. Moreover, as workarounds develop at points of practice where no systematic solution exists, aiming product development activities at these points help close gaps in achieving and maintaining patient safety. This study was not registered.

Comment 2: Introduction: The introduction effectively contextualizes the topic's relevance, supported by 20 references, clearly presents the problem, and concludes with the objective. However, the objective presented in the introduction differs from the one stated in the abstract. The authors should align the objective in the introduction with the one in the abstract. The last two sentences of the final paragraph should be removed from the introduction and included in the materials and methods section.

Response 2: Thank you for catching this inconsistency in our terminology. We have restated the purpose at the end of the introduction to align with the one stated in the abstract. (Page 2, Line 82-84) We also moved the last two sentences of the final paragraph of the introduction into the materials and methods section. (Page 2, Line 86-91)

The purpose of this study was to evaluate a medical device to safely disconnect catheter connections through a user-centered design approach and a socio-technical framework to better understand users’ needs for improving the design.

Comment 3: Materials and Methods: Clarify whether the nurses signed an informed consent form agreeing to participate in the study, as this is not explicitly mentioned. Include the sample size information (currently in the results section) in the methods section and describe how participants were recruited (whether intentional or random), along with the inclusion and exclusion criteria. It is recommended to mention in the methods section that nurses were observed by researchers during task execution, as this detail appears only in the article's strengths and limitations section.

Response 3: We agree this information needs to be included in the methods section. We have added details about recruitment processes, recruitment eligibility and the consent process, recruitment eligibility, sample size, and observations. (Pages 2-3, Lines 98-118)

Participants verbally consented at AMC 1 and were exempt from consent at AMC 2. Participants were first recruited from AMC 1, which resulted in iterative improvements to the design. This was followed by recruitment at AMC 2. See Figure 1 for study schema.

Purposive sampling was used at both AMCs to recruit nurses working in units or clinics that care for patients with catheters. Eligibility at both AMCs was based on nursing qualifications, specifically those trained in catheter management care. Nurses without prior experience caring for patients with catheters were excluded. Participants were contacted through email by our research team for scheduling usability sessions in the nurses’ home unit at each respective institution. To avoid introducing bias or persuasion, there was no direct reporting relationship between the research team and the participants. A total of 139 registered nurses (RNs) across both AMCs engaged in usability testing and asked to complete a previous experience survey after concluding the session. Participants in the usability testing sessions from AMC 2, were also asked to complete a post-usability survey to provide perspectives using the disconnection device. Participants were provided with a QR code to access both the previous experience and post-usability surveys. All surveys were completed within 1 week of the usability testing session. Units included critical care, rehabilitation, adult and pediatric in-patient, blood and marrow transplant, oncology infusion clinics, dialysis, and intensive and progressive care units at both respective AMCs. All participants were observed by researchers while performing the task during the usability testing session. Observational data were documented for qualitative analysis.

Comment 4: Results: The first paragraph of section 3.1 should be moved to the methods section, and the results section should begin with the second paragraph, describing the profile of the studied sample.

Response 4: We agree and have moved the first paragraph to the methods section (see Response 3). The first paragraph of section 3.1, now begins by describing the profile of the studied sample. (Page 9, lines 323-329)

Comment 5: Conclusion: The conclusion should address the research objective. Therefore, the authors should include the following statement: "The nurses who tested the developed medical device reported high acceptability, accessibility, ease of use, and improved task performance."

Response 5: We agree and have added the statement in the conclusion (Page 14, Line 499-501)

Addressing the need for a specialized disconnection device in catheter connection management was achieved through a user-centered design approach. The nurses who tested the developed disconnection device reported high acceptability, accessibility, ease of use, and improved task performance. By applying..

Comment 6: References: Ensure compliance with the journal's requirement regarding the percentage of references published within the last five years, as many of the 37 references cited are older than five years.

Response 6: Thank you for drawing attention to our references. Much of the reason why we did this work is because this particular task has been overlooked in the nursing profession, therefore current literature does not exist. Most of our references pertain to the frameworks and methods we applied, which are more than 5 years old. We have discussed this with the editor as well.

Reviewer 3 Report

Comments and Suggestions for Authors

Thank you for the opportunity to review this manuscript that describes a study employing a user-centered design to iterate a prototype of a new device that facilitates catheter disconnections. This is an important practice area that requires innovation; therefore, this work has the potential to significantly impact the work of nurses and other members of the health care team. The use of a user-centered design and examining the usability of the device within the context of a socio-technical framework is a strong methodological approach for this study. Similarly, the recruitment of nurses and data collection from 2 different academic-medical centers, collecting both qualitative and quantitative data is a strength of this study. Despite these strengths, there are numerous weaknesses in the write-up of this study and therefore, based on my review, significant revisions are required before publication. I am outlining an overview of the areas noted for improvement.

Background: In general, the background is written well and clearly describes the need for this work. Of note, line 63-65 is unclear and would benefit from revisions. This line is also the first time I noted the use of the word “components” which is used multiple times in the paper; however, it is unclear what this word is referring to (perhaps, catheter connection types?).

Purpose: The purpose of this paper is well written and clear

Methods:

·       The recruitment is described as a single event in the methods section; however, that doesn’t align with Figure 1.

·       Figure 1 and the recruitment and data collection section would benefit from further explanation of the participants. For example, were the participants involved in the usability testing the same participants who completed the previous experience survey? And in AMC 2, did the same participants complete the post-survey? What was the timing of the data collection? Did the survey occur immediately before the usability testing, or at a different time? Furthermore, were all “nurses” Registered Nurses?

·       Recruitment details should be described further, for example, how were potential participants contacted and what was the relationship between the research team and the AMCs? Were specific units recruited from?

·       The final paragraph in the study design includes information about data analysis, I recommend that it should be revised to speak to the design elements of how the data will be analyzed.

·       Is there a more specific name that can be used to describe the prototype device? Perhaps “disconnection device”, rather than “medical device”? It is difficult to discern the different types of medical devices referred to in the manuscript and differentiate with this specialized disconnection device.

·       Section 2.2.1.3 would benefit from revision, re-ordering the sentences for better flow.

·       The surveys used for data collection must be described further, for example, who and how were the surveys developed? How many items were on the survey? How were disconnections described (any distinction between unintentional disconnection compared to troubles with manually disconnecting hubs?) The qualitative data collection method requires further description, as described the data appears categorical (i.e. not qualitative data intended for thematic analysis as described).

·       The usability testing process requires additional description. For example, where did the sessions occur? What data were collected and how were they collected? How many sessions occurred and how many were in each session?

·       The multidisciplinary study team is a strength. Please report on the relationship to the AMC and how many participants were in each usability session.   

·       The rationale for only AMC 2 (which is not blinded on page 6, line 195) completing the post-usability survey should be included.

·       Section 2.2.5 is unclear and should be revised. Revisions should clarify how the framework was used to analyze the data, the mapping process, and to be consistent with the other elements of the manuscript.

·       The data analysis section is insufficient to describe how the analysis was conducted. Revisions should focus on clarity, and ensuring that it demonstrates rigor and replicability, as well as consistency with the data collection description. For example, demographic data is included in the analysis section but not included in the section describing data collection. The use of “subjective” assessment reads contrary to the scientific method that is intended to objectively measure, describe, or explain the phenomenon of interest.

Results

·       Without a clear description of data collection and analysis, it is difficult to interpret the results. In general, there are inconsistencies between the data being presented and the findings. For example, the finding on training (p 7 line 287) introduces “adequate” but the data collection was described as dichotomous (yes/no); there is little obvious alignment/connection between the example quotes and the themes provided in the table; the triangulation section is unclear (included Figure 3).

Discussion

·       The discussion should clearly link with the content presented in the manuscript and relate to the study’s purpose, while contextualizing the findings with what is known in the literature. This discussion section seems to extend beyond the findings and introduces new ideas (such as workarounds) that do not seem related to the purpose.

·       The strengths and limitations section includes information that is not described in the paper and the introduction of gemba walks is difficult to follow.

Conclusions: The conclusions do not align with the purpose.

References: I am surprised that the Infusion Nurses Society’s recent practice guidelines are not included in the reference list, but the CDC guidelines are (which have not been maintained). I recommend review of the INS guidelines and integration as appropriate.

General note across the manuscript: There are multiple times where the text has a reference code that was converted to an error in the text. Each of these needs to be resolved.

Again, thank you for the opportunity to review this manuscript. I hope these comments help with revisions to improve the quality of the manuscript and make it suitable for publication.

Author Response

Thank you for your insightful comments and interest in our research. We appreciate the time and effort you have dedicated in providing valuable feedback on our manuscript.

Comment 1 Background: In general, the background is written well and clearly describes the need for this work. Of note, line 63-65 is unclear and would benefit from revisions. This line is also the first time I noted the use of the word “components” which is used multiple times in the paper; however, it is unclear what this word is referring to (perhaps, catheter connection types?).

Response 1: We agree and have made edits to this sentence (page 2, line 67-69). We have also adjusted the use of the term “ catheter components” to “catheter connections” throughout the manuscript.

In recent decades, the luer connection (19) has advanced performance of managing catheters by providing a secure connection among various catheter connection types (e.g., medical tubing, needleless syringes, needleless claves, administration/irrigation bags).

The catheters’ luer connection (19) has advanced the ability to administer treatments by making it easier to connect various components for catheter connection management (e.g., needleless syringes, needleless claves, administration/irrigation bags).

Comment 2: The recruitment is described as a single event in the methods section; however, that doesn’t align with Figure 1.

Response 2: Thank you for bringing this to our attention. We have added the following text (section 2.1, paragraph 1-2, page 3, lines 97-118):

Participants were first recruited from AMC 1, which resulted in iterative improvements to the design. This was followed by recruitment at AMC 2. See Figure 1 for study schema.

Purposive sampling was used at both AMCs to recruit nurses working in units or clinics that care for patients with catheters. Eligibility at both AMCs was based on nursing qualifications, specifically those trained in catheter management care. Nurses without prior experience caring for patients with catheters were excluded. Participants were contacted through email by our research team for scheduling usability sessions in the nurses’ home unit at each respective institution. To avoid introducing bias or persuasion, there was no direct reporting relationship between the research team and the participants. A total of 139 registered nurses (RNs) across both AMCs engaged in usability testing and asked to complete a previous experience survey after concluding the session. Participants in the usability testing sessions from AMC 2, were also asked to complete a post-usability survey to provide perspectives using the disconnection device. Participants were provided with a QR code to access both the previous experience and post-usability surveys. All surveys were completed within 1 week of the usability testing session. Units included critical care, rehabilitation, adult and pediatric in-patient, blood and marrow transplant, oncology infusion clinics, dialysis, and intensive and progressive care units at both respective AMCs. All participants were observed by researchers while performing the task during the usability testing session. Observational data were documented for qualitative analysis.      

Comment 3: Figure 1 and the recruitment and data collection section would benefit from further explanation of the participants. For example, were the participants involved in the usability testing the same participants who completed the previous experience survey? And in AMC 2, did the same participants complete the post-survey? What was the timing of the data collection? Did the survey occur immediately before the usability testing, or at a different time? Furthermore, were all “nurses” Registered Nurses?

Response 3: We agree and have modified the methods section to incorporate more details about the study participants. (see Response 2)

Comment 4: Recruitment details should be described further, for example, how were potential participants contacted and what was the relationship between the research team and the AMCs? Were specific units recruited from?

Response 4: Thank you for making this suggestion. We have added more details to describe our recruitment details and the relationship between our team and the AMCs. (see Response 2)

Comment 5: The final paragraph in the study design includes information about data analysis, I recommend that it should be revised to speak to the design elements of how the data will be analyzed.

Response 5: Thank you for suggesting we revise the last paragraph. We have removed the text from this section, as the design elements of how the data will be analyzed are better explained in section 2.2.5. (Page 7, lines 254-272 and added Table 1. Data collection mapped to Valdez’ socio-technical framework)

Comment 6: Is there a more specific name that can be used to describe the prototype device? Perhaps “disconnection device”, rather than “medical device”? It is difficult to discern the different types of medical devices referred to in the manuscript and differentiate with this specialized disconnection device.

Response 6: We agree and have changed the description of the “medical device” to “disconnection device” throughout the manuscript, including the manuscript title, when referring to our specific device.

Comment 7: Section 2.2.1.3 would benefit from revision, re-ordering the sentences for better flow.

Response 7: Thank you for making this suggestion. We have revised this section for better flow. (section 2.2.1.3, lines 177-187)

To facilitate disconnection, one or two medical devices may be used. Prior to use, the medical device should be wiped with a sanitizing cloth. For disconnection, the user holds the device in either hand, guiding the catheter or catheter connection to the appropriately sized grip area. By squeezing the arms of the device together, the user can firmly grasp the catheter or catheter connection. In instances when additional leverage is needed, two disconnection devices can be used simultaneously. For example, if the first device is applied to the catheter, the second device would be applied to the catheter connection. Leverage is then applied in opposing directions for disconnection.  Once this is achieved, the user relaxes their grip and removes the disconnection device from the male or female part of the catheter or catheter connection. The medical device should be wiped with a sanitizing cloth after each use.

Comment 8: The surveys used for data collection must be described further, for example, who and how were the surveys developed? How many items were on the survey? How were disconnections described (any distinction between unintentional disconnection compared to troubles with manually disconnecting hubs?) The qualitative data collection method requires further description, as described the data appears categorical (i.e. not qualitative data intended for thematic analysis as described).

Response 8: We agree and have expanded on this section to include details about who and how the surveys were developed. We also elaborated on the specific questions asked, to indicate the need for thematic analysis of open-ended questions.

(Section 2.2.2 Previous Experience Survey, page 5, lines 189-207)

Our research team developed the survey questions to validate evidence from our pilot study, which indicated users had challenges with disconnections. All participants were informed that the survey focused solely on experiences with manually disconnecting catheter connections.

Quantitative: Nurses at AMC 1 reported on the frequency in which problems occur with disconnections using a Likert-type scale, with responses ranging from 0 times per month, 1-4 times per month, 5-10 times per month, and more than 10 times per month. Nurses at AMC 1 also reported on whether they have received training related to mitigating problems with disconnecting luer connections, with a yes/no response option.

AMC 1 and AMC 2 nurses’ previous experiences disconnecting luer connections were quantified through three survey questions, including whether they had experienced any of the following, 1) difficulty disconnecting, 2) request for help with disconnecting, and 3) use of an assistive medical device/product to disconnect.

Qualitative: Nurses at AMC 1 reported the types of medical devices/products currently utilized with luer connections and responded to an open-ended question to further elaborate on their previous experiences with luer connections. This open-ended question was only included in the survey from AMC 1, as we recognized this information was verbalized during usability testing sessions, therefore it was removed from the survey to reduce participant survey burden. 

(section 2.2.4 Post Usability Survey, pages 6-7, lines 239-252)

Our research team developed an 11-question survey, including Likert scale and open-ended questions to obtain evidence on the user experience of the specialized disconnection device. Post-usability surveys were only conducted at AMC 2. This contrasts with AMC 1, in which we solely focused on observational data aimed at uncovering innovative design solutions.

Quantitative: Post-usability surveys at AMC 2 were conducted to quantify the usability, length, weight, comfort, grip size, grip function, and acceptability of the disconnection device using a 5-point Likert scale. Acceptability included two questions, “How likely are you to use the device yourself”, and “How likely are you to recommend others use this device.”

Qualitative: Nurses at AMC 2 reported on their experience using the disconnection device after usability testing. Open-ended survey questions included, “What additional features or modifications would enhance the usability of the device?”; “Which aspects of the device functioned well?”; and “ there any concerns with how this device functions?”

Comment 9: The usability testing process requires additional description. For example, where did the sessions occur? What data were collected and how were they collected? How many sessions occurred and how many were in each session?

Response 9: We agree and have added information about where the sessions occurred, what data were collected, how the data were collected, how many sessions occurred, and how many participants were in each session.

(section 2.2.3 Usability Testing, page 7, lines 216-236)

Usability testing sessions were either held in workrooms, training rooms, or conference rooms of each respective unit. When testing sessions were held in “training rooms”, mannequins were used in addition to the transportable kit. Sessions were held in 29 units (AMC 1, N=11; AMC 2, N=18), with participant counts ranging from 2-10 nurses per session.

A multidisciplinary research team including a clinical nurse scientist, a registered nurse, a biomedical engineer, and a human factors expert led usability testing sessions. At AMC 1, sessions were led by our human factors expert and biomedical engineer. At AMC 2, sessions were led by our clinical nurse scientist and registered nurse. Our research team began each usability testing session by providing participants with background information on the development of the medical device. Nurses were not given instructions on how to hold the medical device, as our research team was interested in observing initial responses on how the medical device best fit in their hands, as well as how nurses would use the medical device on various connections. During usability sessions, nurses were asked to perform the task of disconnecting varying luer connections using the medical device. Nurses were also asked to describe their current processes for disconnecting luer connections. We documented both non-verbal (e.g., hand placement, catheter placement and performance success and failures) from observing nurses’ task performance, as well as their verbal feedback on both current processes (e.g., challenges and successes, and workarounds), and use of specialized medical device (e.g., challenges and successes, suggested improvements, and satisfaction). 

Comment 10: The multidisciplinary study team is a strength. Please report on the relationship to the AMC and how many participants were in each usability session.   

Response 10: Thank you for acknowledging the strength of our study team. We have now included the relationship of each team member to the AMC. We have also included the number of participants in each session.

(Section 2.2.3 Usability Testing, page 6, lines 217-222)

When testing sessions were held in “training rooms”, mannequins were used in addition to the transportable kit. Sessions were held in 29 units (AMC 1, N=11; AMC 2, N=18), with participant counts ranging from 2-10 nurses per session.

A multidisciplinary research team including a clinical nurse scientist, a registered nurse, a biomedical engineer, and a human factors expert led usability testing sessions. At AMC 1, sessions were led by our human factors expert and biomedical engineer. At AMC 2, sessions were led by our clinical nurse scientist and registered nurse. Our research team began each usability testing session by providing participants with background

Comment 11: The rationale for only AMC 2 (which is not blinded on page 6, line 195) completing the post-usability survey should be included.

Response 11: Thank you for catching this, we have blinded AMC2. We also provided the rationale. (section 2.2.4 Post Usability Survey, page 6, lines 240-245)

Post-usability surveys were only conducted at AMC 2. This contrasts with AMC 1, in which we solely focused on observational data aimed at uncovering initial design solutions for a high-fidelity prototype of the disconnection device.

Post-usability surveys were conducted at AMC 2 to quantify the usability, length, weight comfort, grip size, grip function, and acceptability of the disconnection device, using a 5-point Likert scale. This contrasts with AMC 1, in which we solely focused on observational data aimed at uncovering innovative design solutions.

Comment 12: Section 2.2.5 is unclear and should be revised. Revisions should clarify how the framework was used to analyze the data, the mapping process, and to be consistent with the other elements of the manuscript.

Response 12: We agree and have added a table that displays how our data collection was initially mapped to Valdez’ framework. (Table 1. Data collection mapped to Valdez' socio-technical framework). We also added text to section, 2.3 Data Analysis, page 8, lines 277-284, to clarify how the data was analyzed.

All data were subjectively assessed, analyzed and quantified from survey responses and usability testing sessions. To reduce unintentional bias of results, further review was conducted by two members of our research team (DRR and LM), who do not have prior experience in performing this task. Data collected from surveys and usability testing sessions were first mapped to Valdez’ framework, as displayed in Table 1, to establish a priori codes. Qualitative data was analyzed as described in Section 2.3.3 and Section 2.3.4 to generate themes. Themes generated from the qualitative data were then combined with the quantitative data to determine how each theme best fit under one of the four domains, while highlighting the interconnections between each of the four domains; user, task, organization and technology.

Comment 13: The data analysis section is insufficient to describe how the analysis was conducted. Revisions should focus on clarity, and ensuring that it demonstrates rigor and replicability, as well as consistency with the data collection description. For example, demographic data is included in the analysis section but not included in the section describing data collection. The use of “subjective” assessment reads contrary to the scientific method that is intended to objectively measure, describe, or explain the phenomenon of interest.

Response 13: Thank you for suggesting we further describe our data analysis. Please see response 12. We have chosen to use the term “subjective” which is often used in qualitative research, as it accurately portrays the insights obtained from the participants and when combined with an established framework, maintains academic rigor. We also added context to the Strengths and Limitations sections:

This study was a subjective assessment of nurses’ experiences; therefore, to mitigate unintentional biases of data collection, analysis and reporting, the usability sessions were performed by differing research team members, surveys were used to collect data, and collective consensus was reached on generated themes and final analysis. Further review of results was conducted by two members of our research team (DRR and LM), who do not have prior experience in performing this task.

Results

Comment 14: Without a clear description of data collection and analysis, it is difficult to interpret the results. In general, there are inconsistencies between the data being presented and the findings. For example, the finding on training (p 9 line 343) introduces “adequate” but the data collection was described as dichotomous (yes/no); there is little obvious alignment/connection between the example quotes and the themes provided in the table; the triangulation section is unclear (included Figure 3).

Response 14: Thank you for noticing this inconsistency. We have removed the word adequate from the results section. Only 23% (n=11/61) of nurses reported receiving adequate training on how to resolve problems with luer connections. (Page 9, line 355) We have also added (Table 1. Data collection mapped to Valdez' socio-technical framework) which displays how our data collection was initially mapped to Valdez’s framework.

We have also added further explanation of how we used Valdez’ framework. (2.3 Data Analysis, page 8, lines 277-284) This should clarify how the themes in Tables 3 and 4 were generated.

All data were subjectively assessed, analyzed and quantified from survey responses and usability testing sessions. To reduce unintentional bias of results, further review was conducted by two members of our research team (DRR and LM), who do not have prior experience in performing this task. Data collected from surveys and usability testing sessions were first mapped to Valdez’ framework, as displayed in Table 1, to establish a priori codes. Qualitative data was analyzed as described in Section 2.3.3 and Section 2.3.4 to generate themes. Themes generated from the qualitative data were then combined with the quantitative data to determine how each theme best fit under one of the four domains, while highlighting the interconnections between each of the four domains; user, task, organization and technology.

2.3.3. Previous Experience Survey - Qualitative

Open-ended survey responses were analyzed using a hybrid approach to thematic analysis, including three phases of analysis in which data were refined to assess the meaningfulness of themes related to nurses’ previous experiences with luer connections (29). Initial responses were first mapped to Valdez’ framework. Then, in phase 1 of the thematic analysis, data from survey responses were categorized by a priori themes based on user-centered design principles, such as current tasks, workflows, usability and functionality. In phase 2, we created the initial a posteriori codes and in phase 3, we combined the a priori and a posteriori codes into family codes to structure the findings from the open-ended survey responses (29). All codes underwent a comprehensive review and were systematically clustered into overarching themes through collaborative discussion and consensus-building among the research team.

Discussion

Comment 15: The discussion should clearly link with the content presented in the manuscript and relate to the study’s purpose, while contextualizing the findings with what is known in the literature. This discussion section seems to extend beyond the findings and introduces new ideas (such as workarounds) that do not seem related to the purpose.

Response 15: Thank you for raising this point. The purpose of this study was to evaluate a disconnection device to safely disconnect catheter connections through a user-centered design approach and a socio-technical framework to better understand users’ needs for improving the design.

We have added this sentence in paragraph 1, lines 415-417, “This study demonstrated how using a socio-technical framework to understand the interconnection between social and technical domains can inform the design and optimize organizational success for addressing the needs of end users.

Utilizing the socio-technical framework allowed us to understand how the interconnection between social and technical domains can inform the design and optimize organizational success to develop a device that meets end users’ needs. This includes how nurses’ tasks (including use of unsafe workarounds) are related to the technology available and the organizational structure. We have added the following statement to clarify the discussion of workarounds.

      (Discussion, page 16, paragraph 2, lines 490-493)

Unfortunately, evidence from this study suggests that medical errors have occurred from these workarounds. For example, hemostats were reported to have caused damage to catheters, resulting in the need for repair or replacement of catheters, impacting patient safety and overall healthcare costs. Currently, no device exists to safely disconnect catheter connections, therefore those performing this task must decide between unsafe workarounds or placing patients at higher risk of infection by not changing catheter connections.

Comment 16: The strengths and limitations section includes information that is not described in the paper and the introduction of gemba walks is difficult to follow.

Response 16: We agree and have modified the strengths and limitations section, including deleting the reference to “gemba walks”.

(Section 4.1, Strengths and Limitations, page 15, lines 487-494)

This study is limited by not observing nurses performing this task in clinical settings with the medical devices and products currently available. We were also limited by not testing the specialized disconnection device in a clinical setting, as the disconnection device had not been registered with the FDA at the time of this study. However, conducting usability testing sessions allowed nurses to discuss utilization of current medical devices and products, and perform the task in a simulated environment without burdening patients. The insights obtained ensured the design of the disconnection device met their needs and supported their current workflows.

Comment 17 Conclusions: The conclusions do not align with the purpose.

Response 17: We agree and have added the following sentence to the conclusion for better alignment with the study purpose. (Page 16, line 503-505)

Addressing the need for a specialized medical device in catheter connection management was achieved through a user-centered design approach. The nurses who tested the developed medical device reported high acceptability, accessibility, ease of use, and improved task performance. By applying a socio-technical framework, we were also able to understand the context around nurses’ performing the task of disconnecting luer connections.

Comment 18 References: I am surprised that the Infusion Nurses Society’s recent practice guidelines are not included in the reference list, but the CDC guidelines are (which have not been maintained). I recommend review of the INS guidelines and integration as appropriate.

Response 18: We agree and have added references to the Infusion Nurses Society guidelines (Reference #11 - Nickel et. al. 2024) in the Introduction section of this manuscript. (Page 2, paragraphs 1-2, lines 49-60)

Comment 19: General note across the manuscript: There are multiple times where the text has a reference code that was converted to an error in the text. Each of these needs to be resolved.

Response 19: Thank you for bringing this to our attention. All references have been fixed.

Round 2

Reviewer 3 Report

Comments and Suggestions for Authors

Thank you for your revisions to this manuscript. Overall, the revisions are satisfactory and improve the clarity of the manuscript. 

There are a few remaining notes to update before publication:

·       Section 2 Material and Methods:

·       The first line is unclear - the use of "early focus" is confusing; revise.

·       The IRB section, page 3 line 97, appears to be missing the word protype: "which resulted in iterative improvements to the prototype design" 

·       2.2 study design - noted "Error! Reference source not found" in lines 124 and 133 (these are not the only instances)

·       The final sentence (p. 3, lines 134 -135) in section 2.2 seems superfluous. "Recruitment stopped..." Additionally, the reference indicates grounded theory, which is not an appropriate method for this work. I also imagine that those details are in the publication of the pilot study referenced. I recommend deletion. 

·       Thank you for the edits from "medical device" to "disconnection device" - that reads much clearer. However, that edit needs to be added to Figure 1 (2 boxes) and Figure 4 (technology box) for consistency. Also, in Figure 1, please change the citation from Cole 2023 to match the formatting for the rest of the paper (i.e., 19). 

·       The data in Table 2 does not match the data presented in the text. Furthermore, the table does not seem to add any new information to the manuscript. I propose you delete Table 2. 

·       Table 3 presents quotes and themes; however, the themes do not align with the representative quotes. The quotes included in the themes "patient safety," "challenges and work arounds", and "undesirable task performance" are not distinguishable from one another. This represents incomplete analysis of the data and diminishes trustworthiness of the data and the findings. Furthermore the "infection control" quotes don't seem to have anything to do with infection control...not without extrapolating from the statements to an assumption of connection with the goal of reducing infection risk. 

·       Of note, Table 4 the link between the themes and quotes are much easier to see. 

·       Results section 3.6 Triangulation; the use of the word coding in line 387 does not align with the way the information in Table 1 is presented. Table 1 indicates how the data were mapped to the socio-technical system, not the codes. This requires clarification.  

Thank you for your ongoing work to improve this manuscript. 

Author Response

Thank you again for taking the time and effort to review our manuscript. We have addressed all comments and believe these improvements will add value for the readers of Nursing Reports.

Comment 1: Section 2 Material and Methods: The first line is unclear - the use of "early focus" is confusing; revise.

Response 1: We agree and have revised this sentence to read, “This study included an early focused on users and their tasks, usability testing to understand users' perceptions, and an iterative design process.

Comment 2: The IRB section, page 3 line 97, appears to be missing the word protype: "which resulted in iterative improvements to the prototype design" 

Response 2: Thank you for noticing that omission. We have now added the word prototype. “Participants were first recruited from AMC 1, which resulted in iterative improvements to the prototype design.”

Comment 3: 2.2 study design - noted "Error! Reference source not found" in lines 124 and 133 (these are not the only instances)

Response 3: Thank you for recognizing that again. We have fixed and double-checked all reference errors.

Comment 4: The final sentence (p. 3, lines 134 -135) in section 2.2 seems superfluous. "Recruitment stopped..." Additionally, the reference indicates grounded theory, which is not an appropriate method for this work. I also imagine that those details are in the publication of the pilot study referenced. I recommend deletion. 

Response 4: We agree and have deleted this statement.

Comment 5: Thank you for the edits from "medical device" to "disconnection device" - that reads much clearer. However, that edit needs to be added to Figure 1 (2 boxes) and Figure 4 (technology box) for consistency. Also, in Figure 1, please change the citation from Cole 2023 to match the formatting for the rest of the paper (i.e., 19). 

Response 5: Thank you for this suggestion. We have revised the text and citation in Figure 1, and the text in Figure 4.

Comment 6: The data in Table 2 does not match the data presented in the text. Furthermore, the table does not seem to add any new information to the manuscript. I propose you delete Table 2. 

Response 6: We agree and have deleted Table 2. We have also deleted the first sentence under 3.2 Previous Experience Survey – Quantitative, which referenced Table 2. (page 9, Line 334)

Comment 7: Table 3 presents quotes and themes; however, the themes do not align with the representative quotes. The quotes included in the themes "patient safety," "challenges and work arounds", and "undesirable task performance" are not distinguishable from one another. This represents incomplete analysis of the data and diminishes trustworthiness of the data and the findings. Furthermore the "infection control" quotes don't seem to have anything to do with infection control...not without extrapolating from the statements to an assumption of connection with the goal of reducing infection risk. Of note, Table 4 the link between the themes and quotes are much easier to see. 

Response 7: Thank you for bringing this to our attention. We have carefully re-evaluated our themes and provided context for each. Additionally, we have combined the themes of patient safety and infection control, and redistributed a few quotes to ensure a more balanced representation across all themes.

Patient safety and infection control focused on i) risk of infection due to improper handling of various catheters, ii) importance of following protocols to prevent infection, and iii) impact on patient safety from using off-label medical devices such as hemostats. Nurses’ time and efficiency focused on time-consuming processes to mitigate this problem. Challenges and workarounds focused on the various methods and devices to mitigate disconnection challenges. Undesirable task performance focused on i) the inconsistency in disconnections, and ii) lack of training. 

Comment 8: Results section 3.6 Triangulation; the use of the word coding in line 387 does not align with the way the information in Table 1 is presented. Table 1 indicates how the data were mapped to the socio-technical system, not the codes. This requires clarification.  

Response 8: We agree and have revised this sentence to “Based on the mapping of data to Valdez' socio-technical framework (Table 1), …” (page 13, line 387)